# Tuning graphitic oxide for initiator- and metal-free aerobic epoxidation of linear alkenes

Samuel Pattisson[1], Ewa Nowicka[1], Upendra N. Gupta[1], Greg Shaw[1], Robert L. Jenkins[1], David J. Morgan[1], David W. Knight[1] & Graham J. Hutchings[1]

Graphitic oxide has potential as a carbocatalyst for a wide range of reactions. Interest in this material has risen enormously due to it being a precursor to graphene via the chemical oxidation of graphite. Despite some studies suggesting that the chosen method of graphite oxidation can influence the physical properties of the graphitic oxide, the preparation method and extent of oxidation remain unresolved for catalytic applications. Here we show that tuning the graphitic oxide surface can be achieved by varying the amount and type of oxidant. The resulting materials differ in level of oxidation, surface oxygen content and functionality. Most importantly, we show that these graphitic oxide materials are active as unique carbocatalysts for low-temperature aerobic epoxidation of linear alkenes in the absence of initiator or metal. An optimum level of oxidation is necessary and materials produced via conventional permanganate-based methods are far from optimal.

[1] Cardiff Catalysis Institute, School of Chemistry, Cardiff University, Main Building, Park Place, Cardiff CF10 3AT, UK. Correspondence and requests for materials should be addressed to G.J.H. (email: hutch@cardiff.ac.uk).

Carbocatalysis has become an increasingly popular area of research in recent years. This has, to some extent, been due to the discovery of graphene[1] and the subsequent study of graphene-related materials[2]. One particularly interesting area is the use of oxygen-functionalized materials such as graphene oxide and graphitic oxide (GO). These materials have become popularized due to their use as a precursor to various graphene materials via the chemical oxidation of graphite[3]. This oxidation supplements the remarkable physical properties of graphene with a degree of functionality on the surface, enabling their use as carbocatalysts. Bielawski and co-workers[4] first demonstrated the ability of graphene oxide and GO to catalyse oxidation and hydration reactions. Since this seminal work, these materials have been used in a wide range of reactions and consequently have been the subject of many extensive reviews[5–8]. However, a detailed study of the effect of the preparation method and hence the degree of surface oxidation remains an uncharted area of study for GO. Therefore, any rationalization of the resulting structure of GO and its catalytic activity is currently missing.

Despite the number of examples of carbocatalysis by graphene and related materials, the aerobic epoxidation of linear alkenes remains unreported. Recently, the oxidation of benzylic and cyclic hydrocarbons has been shown to be catalysed by sulfur, boron and nitrogen-doped graphenes. Styrene was also studied, with lengthy reaction times leading to the formation of epoxide[9,10]. Other N-containing sp$^2$ carbons and onion-like carbons have also been shown to catalyse the epoxidation of trans-stilbene[11] and styrene[12]. The epoxidation of linear alkenes is an important chemical process due to the requirement of epoxides as chemical intermediates. The silver-catalysed production of ethylene oxide from ethene and $O_2$ currently represents the only example of the direct oxidation of an alkene to an epoxide using oxygen as oxidant[13]. Alkenes that contain an allylic hydrogen can only be succesfully oxidized when utilizing expensive and stoichiometric oxidants such as hydrogen peroxide. We have previously shown that the more favourable use of atmospheric oxygen as oxidant can be facilitated by gold catalysts along with the addition of catalytic amounts of radical initiators such as azo-bis-isobutyronitrile and tert-butyl hydroperoxide[14]. Both commercial cyclic and linear internal alkenes were shown to contain stabilizers. On removal of these, cyclic alkenes could be epoxidized in the absence of radical initiators[15]. However, initiators are required for the epoxidation of terminal linear alkenes[16]. Hence, apart from the special case of ethene, the epoxidation of terminal alkenes with oxygen has not yet been achieved. Here we show that the low-temperature epoxidation of linear alkenes (oct-1-ene, dec-1-ene and dodec-1-ene) in the absence of radical initiators using a metal-free GO catalyst and atmospheric oxygen as the sole oxidant is possible. The activity of the catalysts for this epoxidation reaction is shown to be highly dependent on the amount and type of oxidant used in the preparation of GO. The conventional permanganate-based Hummers (HU) preparation method[17] for the bulk production of highly oxidized GO is, in this case, found to be inferior to the less common chlorate-based Hofmann (HO) method[18]. The effects of oxidant and level of oxidation on the physical properties

of the final graphenes have been studied previously[19–21]; however, we consider this to be the first instance where the catalytic applications have been shown to be markedly affected by the chosen method of oxidation. Recent work by Nishina et al. has also demonstrated that properties of the final graphene oxides can be affected depending on whether oxygen content is tailored by oxidation of graphite or reduction of highly oxidized graphene oxide[22]. This, along with the current study, should encourage focussing attention on optimizing the preparation method of GO for specific applications.

## Results

**Initial results.** Initial studies were conducted using a partially oxidized GO prepared using a modified HO method, in which potassium chlorate (20 g per 5 g of graphite) was used (GO-HO20) in place of the standard 55 g per 5 g of graphite (GO-HO55) that is used in most preparations. We initially intended employing this material as a support following previous work on carbon-supported gold catalysts; however, this GO was found to be active in the absence of metal or radical initiators for the epoxidation of dec-1-ene (dec-1-ene 10 ml, 90 °C, GO-HO20 0.1 g) (Fig. 1). Reactions conducted in the absence of oxygen resulted in no conversion, suggesting that activity for dec-1-ene epoxidation was a result of catalysis with the activation of molecular oxygen rather than the stoichiometric use of surface oxygen. Time online studies (Fig. 2) also showed that the catalyst was active over long periods of time leading to oxygenated products far in excess of what would be expected from the stoichiometric use of surface oxygen. Therefore, we decided to extend this study to a range of GOs produced by modified HO and HU methods in order to assess the effects of amount and type of oxidant. Schwartz et al.[23] studied the effect of oxygen coverage on the activity of oxygen-functionalized graphenes for the oxidative dehydrogenation of isobutane. This was achieved by the production of highly oxidized GO by the HU oxidation method, followed by partial thermal reduction of the surface. These authors concluded that activity was independent of oxygen coverage and more likely to be dictated by the sp$^2$ to sp$^3$ ratio of the surface carbon. In the present study, a range of GOs were prepared using increasing amounts of oxidant. For the HO method, the oxidant mass ranged from 1 to 55 g of potassium

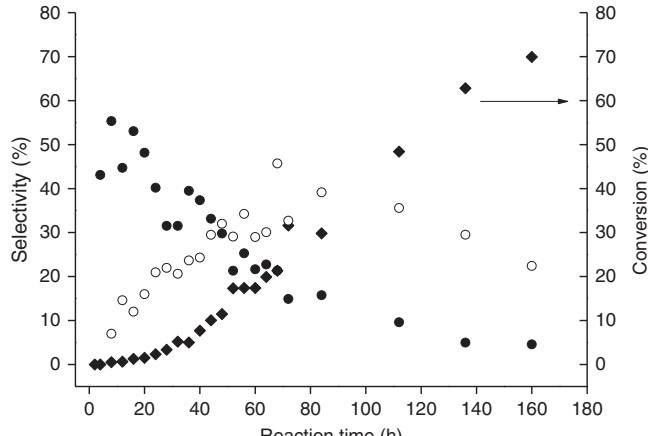

**Figure 2 | Time online studies for the metal and initiator free epoxidation of a linear alkene using GO as a carbocatalyst.** Conversion (black diamond) and selectivity towards epoxide (empty circle) and allylic products (black circle) (dec-1-ene 10 ml, 90 °C, GO 0.1 g). The continued oxidation over longer time frames is indicative of catalysis rather than stoichiometric use of surface oxygen. Allylic oxidation diminishes with time as the epoxide becomes the major product.

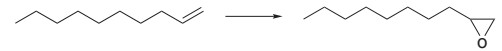

**Figure 1 | Epoxidation of a terminal linear alkene.** (48 h, dec-1-ene 10 ml, 90 °C, GO 0.1 g). The epoxidation of a terminal linear alkene such as dec-1-ene usually requires the use of stoichiometric oxidants, supported metal catalysts and/or radical initiators.

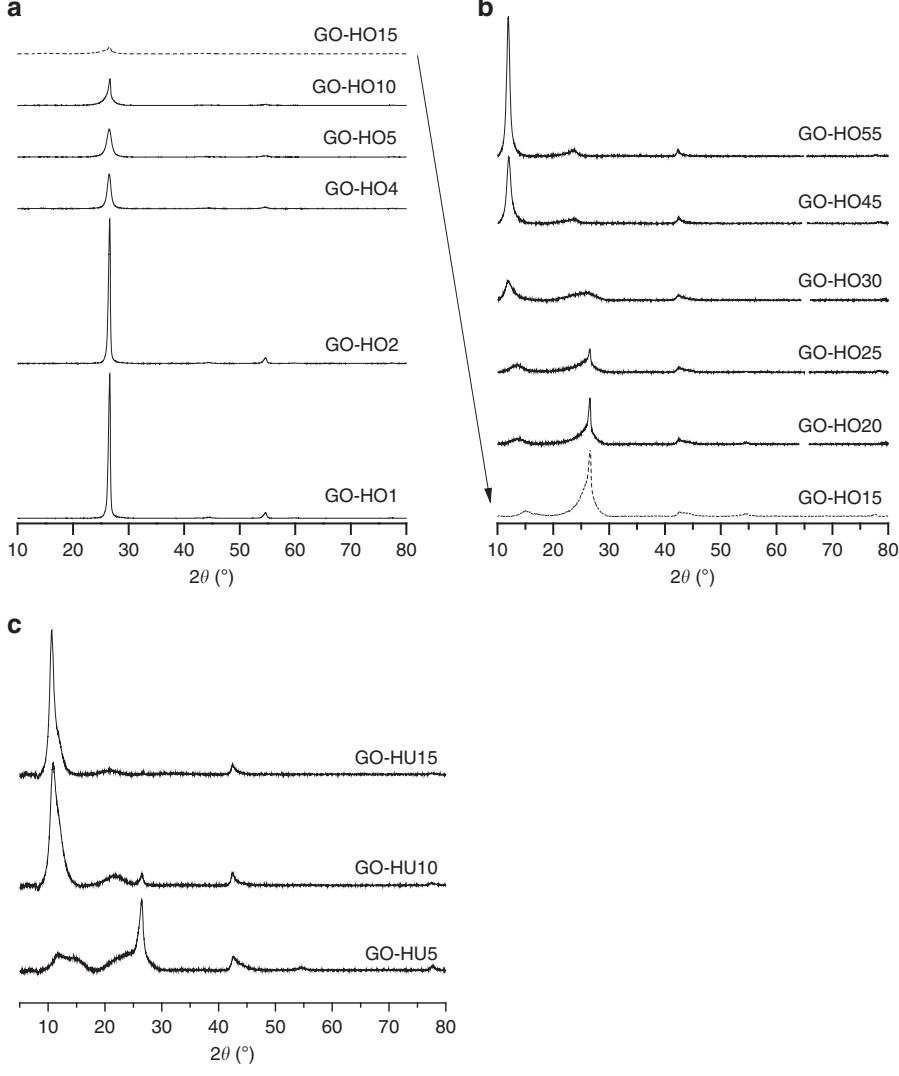

**Figure 3 | XRD analysis of graphitic oxide.** (**a**) Low and (**b**) highly oxidized graphite samples by HO and (**c**) HU method. Chemical oxidation of graphite utilizing increasing amounts of either potassium chlorate or permanganate as oxidant leads to a de-graphitization of the surface leading to loss of crystallinity and the formation of a more amorphous GO-like material.

chlorate per 5 g of graphite and for the HU method, 5 to 15 g of potassium permanganate per 5 g of graphite. The GOs were characterized by X-ray photoelectron spectroscopy (XPS), X-ray diffraction (XRD) and thermogravimetric analysis (TGA) to determine if a structure-activity relationship could be observed for these GO materials for the epoxidation of terminal alkenes so that the origin of this catalysis could be determined.

**XRD analysis**. Powder XRD analysis was conducted on all HO and HU samples to assess the crystallinity and degree of oxidation. The XRD of low-oxidized HO samples GO-HO1 to GO-HO15 (Fig. 3a) shows the gradual oxidation of the graphite surface with the increase in oxidant amount. This results in the gradual break-up of the lattice and loss of crystallinity demonstrated by an overall decrease in the intensity of the 002 peak at 26.5°. The 101 and 004 reflections at 45° and 55°, respectively, also decrease in intensity, showing the gradual de-graphitization of the material. A further increase in the amount of oxidant results in switching from highly crystalline graphitic-like materials to GO-type materials in which the principal reflection is at ~11°, representative of the 002 plane and an increased d-spacing from 3.5 to 7 Å shown in the XRD patterns of

GO-HO15 to GO-HO55 (Fig. 3b). XRD of HU samples GO-HU5, GO-HU10 and GO-HU15 (Fig. 3c) shows a high degree of oxidation and amorphous character similar to the highly oxidized HO samples. The disappearance of the 002 reflection of graphite on oxidation and emergence of that relating to GO is in agreement with results published by Kim *et al.* on the effect of oxidation on graphene oxide structure[20].

**XPS analysis**. The functionality and degree of surface oxidation of each of the samples were analysed by XPS. In graphite and low-oxidized HO samples such as GO-HO1 (Fig. 4a), the principal peak observed at ~284.6 eV is representative of the C–C bonding of sp[2] hybridized carbons. The use of higher amounts of oxidant results in increased lattice oxidation and consequently a decrease in sp[2] character. Therefore, from GO-HO2 to GO-HO55 a reduction in the C–C peak is observed. This is accompanied by the development of a peak at 286.7 eV, which represents hydroxyl functionality. Furthermore, highly oxidized samples show the presence of various peaks above 287 eV, which suggest the presence of epoxy, carbonyl and carboxylic groups, although in HO samples these are limited to small amounts of carboxylic and epoxy groups. Carboxylic functionality is more easily observed in

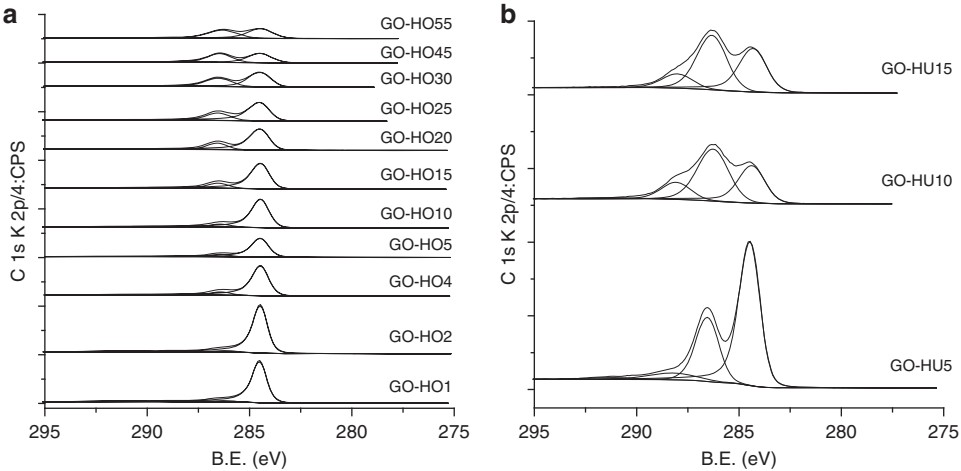

**Figure 4 | XPS analysis of increasingly oxidized graphite.** (**a**) HO and (**b**) HU method. The increasing level of oxidation of graphite is represented by the loss of C–C character and the introduction of oxygen functionality. HU materials show a higher degree of carboxylic functionality compared to HO catalysts due to the increased lattice break-up and subsequent oxidation of edge sites.

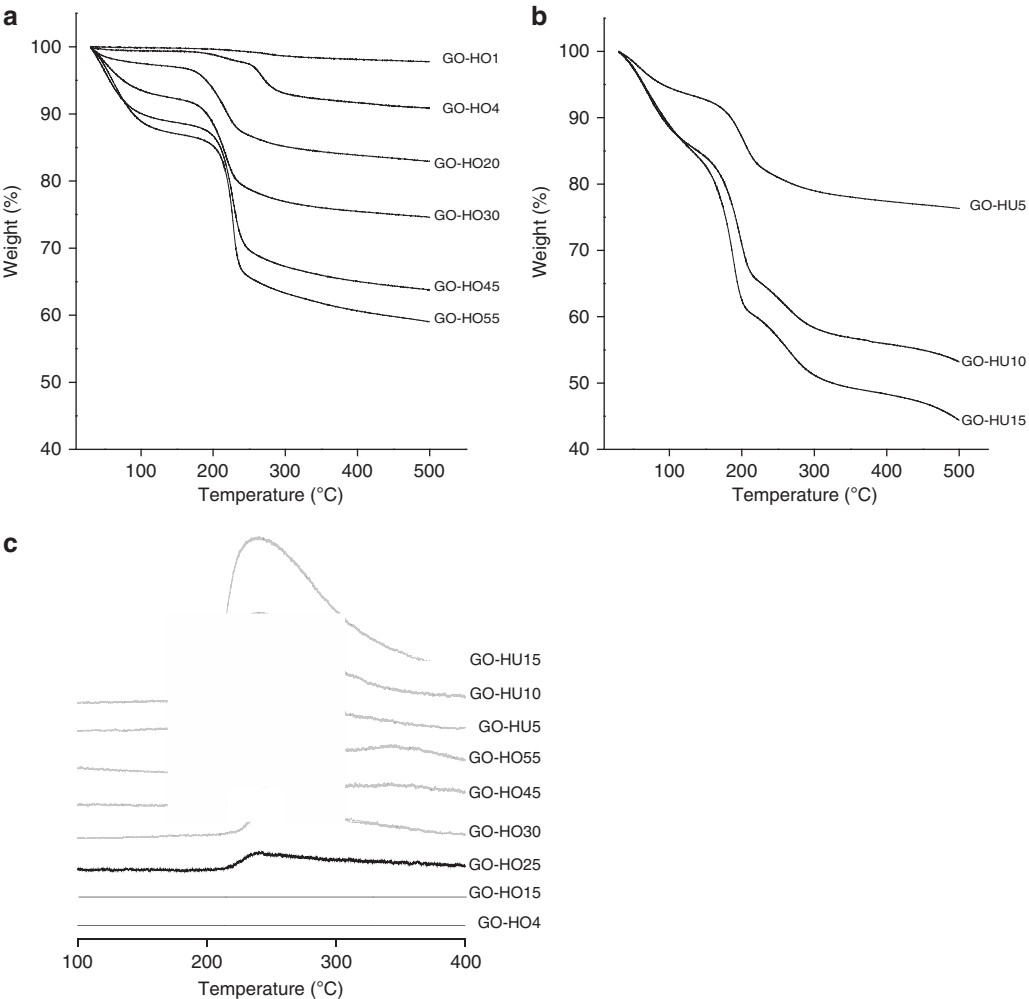

**Figure 5 | TGA of graphitic oxide.** GO prepared by (**a**) HO and (**b**) HU method using increasing amounts of oxidant. (Conditions: 30–500 °C, rate: 5 °C min$^{-1}$, $N_2$ flow; 20 ml min$^{-1}$, GO mass 10 mg) (**c**) TGA–MS, Mz 64 ($SO_2$). TGA of a range of oxidized GOs produces increasing weight losses of $H_2O$, CO and $CO_2$. Materials prepared by HU method show a higher weight loss associated with organosulfates than HO catalysts that are tuned to have similar oxygen compositions.

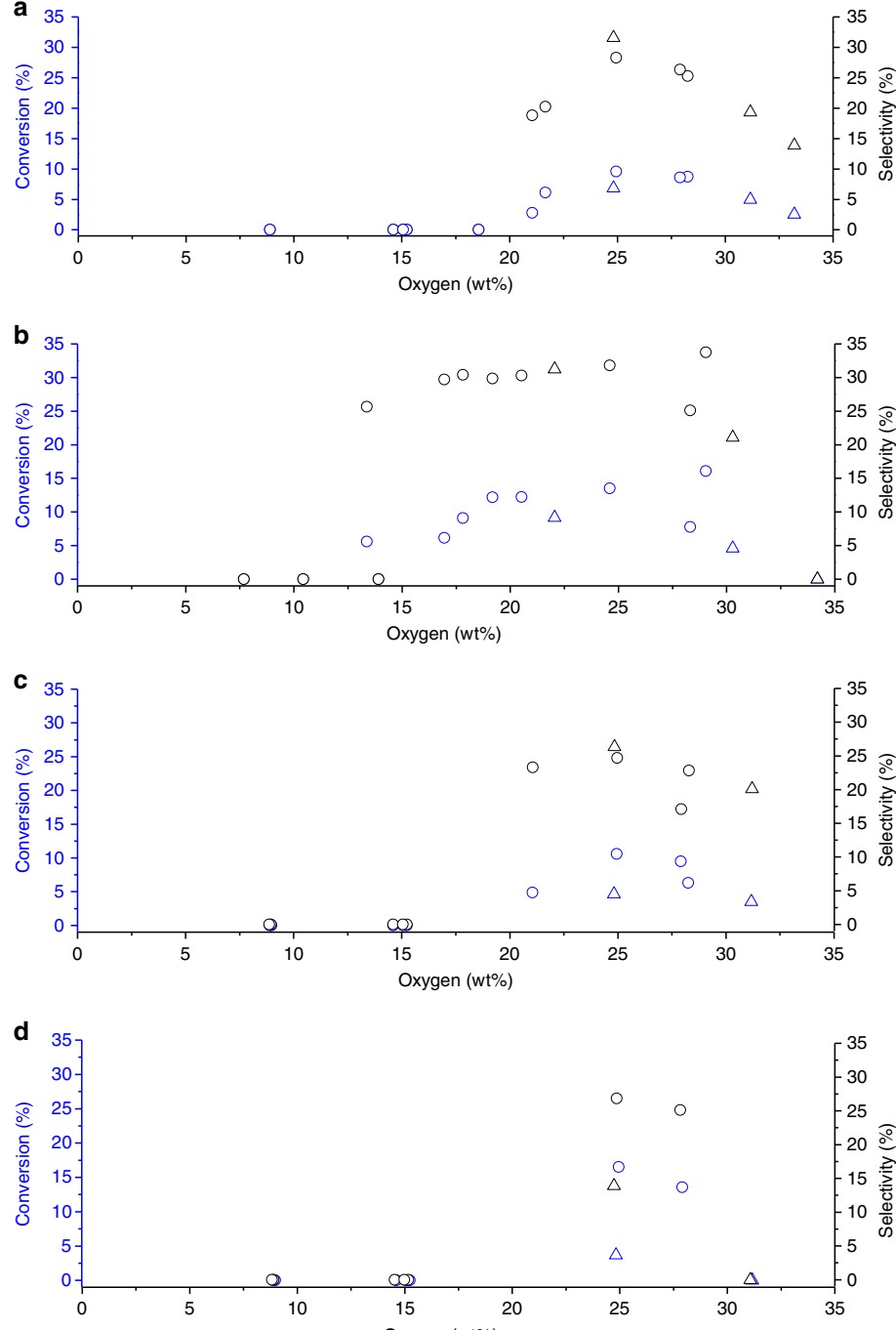

**Figure 6 | Activity of graphitic oxide for the epoxidation of terminal linear alkenes.** (**a**) Unwashed and (**b**) washed GO samples for the epoxidation of dec-1-ene (48 h, dec-1-ene 10 ml, 90 °C, GO 0.1 g). Unwashed samples for the epoxidation of (**c**) dodec-1-ene (48 h, dodec-1-ene 10 ml, 90 °C, GO 0.1 g) and (**d**) oct-1-ene (72 h, oct-1-ene 10 ml, 80 °C, GO 0.1 g). Samples prepared by HO method: ○, HU method: △. The epoxidation of linear alkenes by GO is dictated by an apparent minimum level of surface oxygen and also an optimum, above which, activity decreases. Activity is also sensitive to the presence of both inorganic and organosulfates with the former being removed by additional washing and the latter explain the discrepancy between HU and HO catalysts.

highly oxidized HU samples (Fig. 4b) at ~289 eV. These results are in agreement with previous investigations carried by Pumera *et al.*, who described the higher carboxylic functionality at edge sites in GO-HU's compared to GO-HO's, which contain predominantly hydroxyl and epoxy groups[24].

**TGA analysis**. TGA was conducted on the full range of GO-HO (Fig. 5a) and GO-HU (Fig. 5b) catalysts in order to assess their thermal decomposition under a nitrogen atmosphere over time.

Here only few data examples are chosen for presentation clarity. Low-oxidized GO-HO1 and GO-HO2 demonstrated low weight loss at ~250 °C, most likely due to the loss of lattice water. In GO-HO4, GO-HO5 and GO-HO10, this weight loss at 250 °C is preceded by loss of carbon dioxide, carbon monoxide and lower-temperature release of water. Weight loss due to these molecules is more apparent in GO-HO15 to GO-HO25. In highly oxidized materials such as GO-HO30 to GO-HO55 and all GO-HU materials, a significant weight loss occurs between 50 and 120 °C,

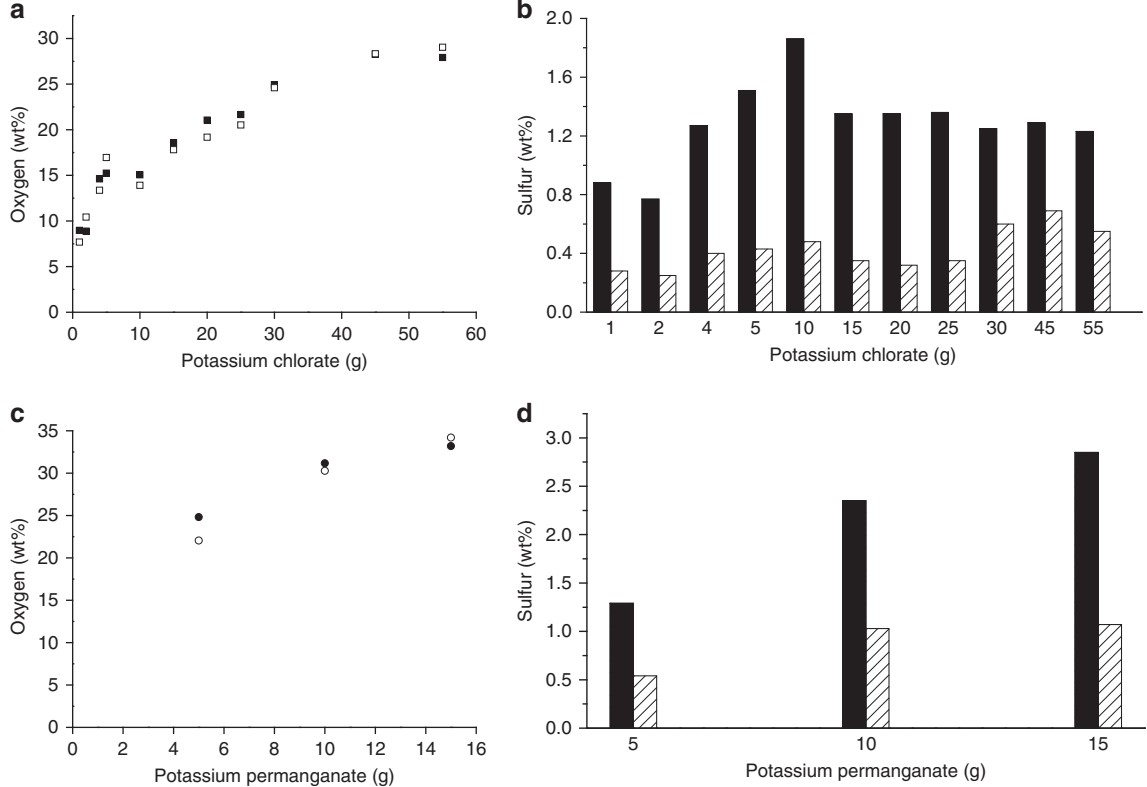

**Figure 7 | Level of oxygen and sulfur determined by XPS analysis before and after additional washing step.** Before (black symbols) and after (empty symbols) graphite oxide washing with water. Samples prepared by HO (**a,b**) and HU (**c,d**) methods. The non-linear relationship between the amount of oxidant and the level of oxygen demonstrates the need for tuning of GO for specific applications. Additional washing leads to the significant reduction of sulfur in HO catalysts, however, less so in HU catalysts due to the retention of covalently bound organosulfates.

most likely due to the loss of physisorbed water. This is followed by further weight loss up to 200 °C due to formation of carbon monoxide and carbon dioxide plus loss of residual water. In highly oxidized GO-HU5, GO-HU10 and GO-HU15 samples, a weight loss between 200 and 250 °C can be explained following the work of Eigler *et al.*[25]. The authors demonstrated the loss of covalently bound organosulfate groups and formation of $SO_2$ accompanied by small amounts of water, carbon monoxide and carbon dioxide. Thermal gravimetric analysis coupled with mass spectrometry (TGA–MS) analysis of our samples showed a direct relationship between the extent of oxidation and the presence of organosulfate groups bound to the GO surface (Fig. 5c). HO catalysts prepared using < 20 g of potassium chlorate were free from organosulfates. Above this level, organosulfates were shown to be present in all catalysts, the level of which roughly follows level of oxidation. Organosulfate was shown to be present in all HU catalysts. Furthermore, HU samples displayed a much higher level of organosulfate even when tuned to the same level of oxygen as HO catalysts. This is viewed most clearly in GO-HU5, which, despite being analogous in level of oxygen to GO-HO25, contains similar levels of organosulfate to GO-HO55, the most highly oxidized HO catalyst.

**Epoxidation of linear alkenes**. The GO materials were tested for the solvent-free epoxidation of dec-1-ene in the absence of radical initiators (Fig. 6a). It was found that a critical amount of surface oxygen (∼ 20 wt%) was required for a GO to be active in the epoxidation reaction. This was achieved using 20 g of potassium chlorate per 5 g of graphite. When lower quantities of oxidant were used, no activity was observed. Furthermore, activity was

shown to be at a maximum at an oxygen concentration of 25 wt%, achieved with 30 g of potassium chlorate as oxidant. Above this level, activity for the epoxidation of dec-1-ene decreased, with highly oxidized HU samples displaying little or no activity. Interestingly, HU samples with oxygen coverage of 25 wt% were less active than HO samples with an analogous amount of oxygen.

One of the major findings of this study was the deleterious effect of low amounts of sulfur for this reaction. Sulfuric acid is essential to the chemical oxidation of graphite due to its oxidizing and intercalating nature[26]. However, the latter means it is also difficult to remove sulfur completely by washing with water. XPS and TGA–MS analysis showed that sulfur was present in all GO samples. The combination of residual sulfur ions and covalently bound organosulfate resulted in the sulfur content in the GOs ranging from ∼ 1 to 2 wt% (Fig. 7b,d). Subjecting samples to an additional washing step resulted in the removal of the majority of residual sulfur ions as confirmed by ion chromatography of the washings and XPS of the dried catalyst. This additional washing step had a significant effect on the activity of the low-oxidized GO samples, as catalysts with as little as 15 wt% oxygen showing activity for the epoxidation of dec-1-ene (Fig. 6b). Only 4 g of potassium chlorate was required to achieve this level of oxygen coverage, which has major implications in the design of GO catalysts, when compared with the standard method of preparation used in previous literature that required 55 g per 5 g of graphite. The highest activity both in unwashed and washed samples was seen at 25 wt% oxygen. Interestingly, highly oxidized HU and HO species showed no increase in conversion even after washing, suggesting that the correlation between oxidation and activity is not linear. It was also observed that increasing the

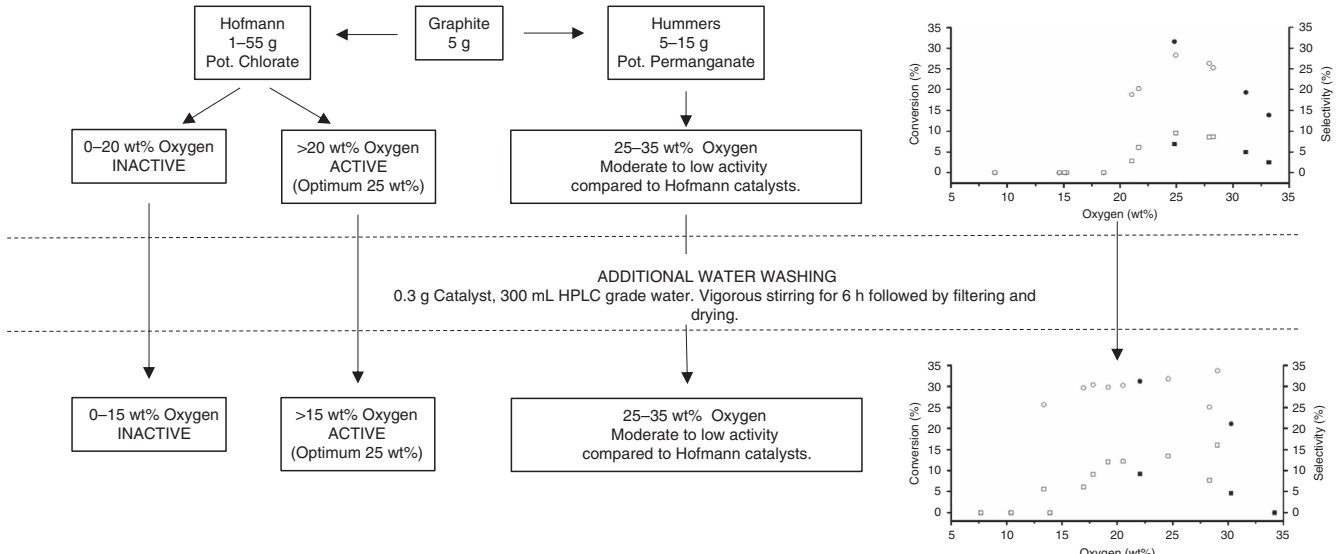

**Figure 8 | Flowchart representation of GO prepared by HO and HU methods.** Activities of a range of GOs before and after the additional washing step.

amount of permanganate from 10 to 15 g had little effect on the level of surface oxygen (Fig. 7c), but increased the level of organosulfates (Fig. 5c), explaining the lower activity of the higher-oxidized material.

Unwashed catalysts were also tested for the epoxidation of dodec-1-ene and oct-1-ene (Fig. 6c,d, respectively). Activities followed a similar trend to that in dec-1-ene epoxidation, where an optimum level of oxygen was observed at 25 wt%. Above this level, both HO and HU catalysts decrease in activity. In both cases, HU catalysts tuned to the same level of oxygen as HO were lower in activity. On the basis of these data, we propose that overoxidation of surface carbon leads to loss of activity, caused by the formation of covalently bound organosulfates, which have a major effect on catalysis despite being present in relatively low amounts. The lower activity of HU catalysts, even after additional washing, can be explained by the higher level of covalently bound sulfates compared to analogous HO catalysts.

A major implication of our findings is that HU samples, commonly used due to the increased level of oxidation and improved safety of the method, are not necessarily the optimum materials for all applications. Certainly for this reaction and similar oxidation reactions, the production of lower-oxidized GOs or the removal of organosulfate by washing with base may be preferred. This is in contradiction to the many examples, where these sulfur groups aid reaction such as in the acid-catalysed ring opening of epoxides[27]. Limitations of chemical oxidation methods stem from the liberation of toxic and explosive gases. However, the synthesis of active materials through use of low quantities of oxidants in both HO and HU methods suggests that these preparation methods could become more accessible to a wider circle of applications. The removal of both sulfur ions and organosulfates by extensive washing in basic solutions could provide active epoxidation catalysts. Alternatively, graphene oxides prepared by sulfur-free methods such as the pyrolysis of sugars[28] or microwave-assisted techniques[29] would completely avoid this poison and offer a route to cleaner epoxidation catalysts, providing the required level of surface oxygen can be obtained.

In summary, the solvent and initiator-free aerobic epoxidation of linear alkenes has been facilitated using GO as a metal-free catalyst. An optimum level of surface oxidation of ∼ 25 wt% was found for the production of an active catalyst. This dependency is not necessarily linearly linked with oxygen weight percent, rather

the extent of oxidation and incorporation of inorganic and organosulfate groups. However, a minimum level of 15 wt% oxygen is required. Removal of inorganic sulfur via an additional washing step enabled the activation of the graphites, even for samples prepared with as little as 4 g of potassium chlorate, which is significantly lower than the literature standard of 55 g per 5 g of graphite (Fig. 8). This becomes significant when considering the hazards involved in the use of large amounts of oxidant. HU samples displayed lower activities than HO catalysts even when tuned to obtain analogous amounts of oxygen as the optimum HO catalyst. These lower-active HU samples contain a much higher level of organosulfates, which remain after additional water washings, as described by TGA–MS.

We suggest that the most commonly used highly oxidized HU catalysts, as obtained by chemical oxidation, may not be the optimum material for all applications and that tailoring of the level of oxidation may be needed. Sulfur acts as a deactivator in the formation of active catalyst for the epoxidation reaction both in the form of inorganic and organosulfate; however, it is unavoidable in current chemical oxidation methods without further treatment of the obtained materials. Recent developments in sulfur-free methods for producing GO such as the pyrolysis of sugars may hold the key to the production of highly active and clean epoxidation catalysts. Removal of all sulfur groups should be facilitated by additional base washing to yield highly pure and active catalysts.

## Methods

**Preparation of GO by modified HO method.** GO was prepared from graphite, according to the reported HO method[19]. Graphite (<20 μm, Sigma-Aldrich) was added to a mixture of concentrated sulfuric (75 ml) and nitric acid (25 ml),which was allowed to cool to 10 °C in an ice bath. Potassium chlorate (1–55 g) was added stepwise to the mixture over a period of 30 min with vigorous stirring. Stirring was continued for 14 h after which the mixture was left in air for 96 h, followed by repeated decantation and centrifugation of remaining GO material.

**Preparation of GO by modified HU methods.** GO was prepared according to previously reported HU method[19] using graphite (<20 μm) as a precursor. Graphite (5 g) was added to a mixture of concentrated sulfuric (87.5 ml) and nitric acid (27.5 ml) under vigorous stirring. Potassium permanganate (5–15 g) was added stepwise over a period of 2 h. The mixture was then allowed to reach room temperature over a period of 4 h, followed by heating to 35 °C for 30 min. Deionized water (250 ml) was added, causing the temperature to rise to 70 °C. A further portion of deionized water (1 l) was added, followed by addition of 3% $H_2O_2$ for the removal of any residual potassium permanganate. The mixture was

allowed to settle overnight after which the GO was separated and washed repeatedly via centrifugation.

**Centrifugation.** Centrifugation was conducted (14,000 r.p.m., 30 min, 20 °C) using a Beckman coulter centrifuge, JLA.16.250 rotor. Samples were dispersed in deionized water (200 ml) before centrifugation. This was repeated until a neutral pH was obtained after which the final supernatant was decanted and the retained HU and HO samples were dried in a vacuum (20 °C) or a regular oven (110 °C), respectively.

**Additional washing procedure and ion chromatography.** GO was subjected to an additional washing for the removal of sulfur. GO (0.3 g) was washed with high performance liquid chromatography (HPLC) grade water (300 ml) under vigorous stirring for 6 h. After this period, a sample of the washing solution was analysed by ion chromatography using a Thermo Scientific Dionex system. GO was dried as described above.

**Epoxidation of linear alkenes.** Dec-1-ene and dodec-1-ene reactions were conducted in a round-bottomed flask (50 ml) fitted with a reflux condenser. The flask was charged with alkene (10 ml) and GO catalyst (0.1 g) after which the mixture was heated to 90 °C and stirred for 48 h utilizing atmospheric oxygen as the oxidant. Oct-1-ene reactions were conducted in a Colaver reactor (50 ml) pressurized with oxygen (3 bar). The amount of substrate and catalyst was concurrent with the above reactions. Reactions proceeded at 80 °C for 72 h. After the reaction, the mixture was centrifuged and analysed by gas chromatography. Major products were concurrent with those identified in previous studies on gold catalysed epoxidation of dec-1-ene[16]. The major product in all cases was the epoxide. Using dec-1-ene as an example: allylic products such as dec-1-en-3-ol, dec-1-en-3-one and dec-2-en-1-ol along with 1,2-decanediol, cracked acid and aldehydes made up the remainder of the product stream. Mesitylene was used as an external standard and an averaged response factor applied to unknown products, however, these accounted for < 10% of products.

**X-ray photoelectron spectroscopy.** XPS was carried out using a Kratos Axis Ultra DLD system and according to the following method. A monochromatic Al Kα X-ray source was operating at 120 W. Data were collected with pass energies of 160 eV for survey spectra, and 40 eV for the high-resolution scans. The system was operated in the hybrid mode, using a combination of magnetic immersion and electrostatic lenses, and acquired over an area $\sim 300 \times 700\,\mu m^2$. A magnetically confined charge compensation system was used to minimize charging of the sample surface, and all spectra were taken with a 90° take-off angle. A base pressure of $\sim 1 \times 10^{-9}$ torr was maintained during collection of the spectra. Binding energies were calibrated using the C1s binding energy of carbon taken as 284.7 eV.

**Powder X-ray diffraction.** XRD analysis was conducted using a PANalytical X'pert pro diffractometer using a Cu Kα X-ray source. Typical scans ranged $2\theta$ from 10 to 80° at 40 kV and 40 mA, although some wide scans were conducted for higher-oxidized samples below 10°.

**TGA analysis.** TGA analysis was carried out using a Perkin Elmer TGA 4000. Standard conditions were consistent with those recommended in the literature[25]. GO samples were typically heated in the temperature range 30–500 °C, with a ramp rate of 5 °C min$^{-1}$ using a N$_2$ flow of 20 ml min$^{-1}$. Approximately 10 mg of catalyst was used for highly oxidized sample and 50 mg for low-oxidized samples.

**TGA coupled with mass spectrometry.** Hyphenated TGA–MS was run on a Pyris 1 TGA linked to a Perkin Elmer Clarus 580 gas chromatography mass spectrometer (GC-MS) using a TL-9000 interface. TGA was performed under helium with temperature ranging from 30 to 500 °C (5 °C min$^{-1}$), using c.a. 10 mg sample in each experiment. The GC–MS was set up, to negate the GC column, with the effluent gas analysed by MS, m/z 18 (H$_2$O), 44 (CO$_2$) and 64 (SO$_2$).

**Data availability.** All data are available from the authors upon reasonable request.

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

## Acknowledgements

The authors wish to acknowledge the financial support of the ERC (ERC-2011-ADG—Grant Agreement N° 291319—Acronym 'AFTERTHEGOLDRUSH') and the EPSRC (Grant EP/L027240/1).

## Author contributions

S.P. and U.G. prepared, tested catalysts and designed the initial experiments. E.N. provided assistance with experimental design and further experiments. D.M. helped with XPS analysis. D.W.K. provided mechanistic insights into the chemistry. Detailed analysis was provided by R.L.J. and G.S. G.J.H. directed the research and all authors contributed to the analysis of the data and the writing of the manuscript.

## Additional information

**Competing financial interests:** The authors declare no competing financial interests.

