## [Peer review file · Nature Communications]

Reviewers' comments:

Reviewer #1 (Remarks to the Author):

In their manuscript, Hutchings and co-workers describe how varying the oxygen content of graphite oxide influences the ability of the material to epoxidize alkenes. Graphite was first treated with various quantities of either potassium chlorate (GO-HO##) or potassium permanganate (GO-HU##) and then the resulting products were characterized using a range of techniques, including XPS, TGA, and MS. In short, it was determined that GO prepared by treating 5 g of graphite with 20 g of potassium chlorate displayed the highest epoxidation activity among of all of the catalysts studied. It was also determined that removal of the residual organosulfates (which, presumably, originate from the sulfuric acid solvent) also improved catalytic activity.

Overall, the work described in the paper is off to an excellent start. The systematic oxidation of graphite is poised to guide future efforts in the rapidly growing area of carbocatalysis and an ability to selectively epoxide terminal olefins without the aid of transition metals will be of intense interest to a broad spectrum of synthetic chemists & engineers. However, additional work is needed before this paper can be considered for publication:

Major Issues

(1) The work described is not "[Lines 80-82] the first instance where the catalytic applications have been shown to be markedly affected by the chosen method of oxidation"; see: Nishina et. al, Sci. Rep. 2016, 6, 21715 (doi:10.1038/srep21715). The introduction should be re-framed.

(2) It appears that the epoxide products were characterized using gas chromatography. Were standards used to guide compound identification? The text indicates that multiple products form during the reaction (what is an "allylic product"?) and thus gas chromatography is insufficient because potential isomers may not be resolved. A secondary technique, such as NMR spectroscopy, should be employed to support the conclusions. Also, how were the gas chromatography signals quantified? Correction factors and/or calibration curves are typically required.

(3) Since XPS data are not necessarily representative of the bulk sample, the GO products should also be analyzed using elemental analysis.

(4) Figure 1: Which catalyst was used to collect these data? Where are the data for the other catalysts? A table that summarizes key results would be informative.

Minor Issues

(5) Line 108: "thermogravimetric" is spelled incorrectly.

(6) Many sections of the Results and Discussion section are obfuscated and long. The paragraph that begins on row 163 is 50 lines long! I suggest that the authors distill and clarify the content of their manuscript so that it is more accessible to a general audience.

Reviewer #2 (Remarks to the Author):

The paper by Hutchings and coworkers is well-written and of great impact. They have demonstrated that GO materials are active catalysts for aerobic epoxidation of aliphatic alkenes and elegantly unraveled details of the optimum level of carbocatalyst oxidation. Their finding that oxidation level typically obtained in the permanganate-based method is far from optimal will have impact on graphene preparation and this result may also open up new promising areas for oxidation of chemical feedstocks. I only have a few minor comments:

1. A reaction scheme showing the epoxidation performed in the study should be added.
2. A graphic showing sequential washing steps and the relative catalyst performance should be added.
3. Epoxidation conditions should be added in the discussion in the main manuscript.
4. A relevant paper on nitronium ion-enabled synthesis of graphene materials should be cited (JACS 2012 134 5850).
5. What is the catalyst performance in styrene or allylic alcohol epoxidation?
6. Can the authors comment if they have tried the same approach for hydrocarbon, alcohol or amine oxidation?

All in all, this is a good piece of paper and I strongly recommend publication.

NCOMMS-16-11308-T Response to referees.

The authors would like to thank the referees for their comments and constructive criticism of the work. The reply to each comment is set out in the following document.

Response to reviewer #1

In their manuscript, Hutchings and co-workers describe how varying the oxygen content of graphite oxide influences the ability of the material to epoxidize alkenes. Graphite was first treated with various quantities of either potassium chlorate (GO-HO##) or potassium permanganate (GO-HU##) and then the resulting products were characterized using a range of techniques, including XPS, TGA, and MS. In short, it was determined that GO prepared by treating 5 g of graphite with 20 g of potassium chlorate displayed the highest epoxidation activity among of all of the catalysts studied. It was also determined that removal of the residual organosulfates (which, presumably, originate from the sulfuric acid solvent) also improved catalytic activity.

Overall, the work described in the paper is off to an excellent start. The systematic oxidation of graphite is poised to guide future efforts in the rapidly growing area of carbocatalysis and an ability to selectively epoxide terminal olefins without the aid of transition metals will be of intense interest to a broad spectrum of synthetic chemists & engineers. However, additional work is needed before this paper can be considered for publication.

Major Issues

(1) The work described is not "[Lines 80-82] the first instance where the catalytic applications have been shown to be markedly affected by the chosen method of oxidation"; see: Nishina et. al, Sci. Rep. 2016, 6, 21715 (doi:10.1038/srep21715). The introduction should be re-framed.

Response

The paper in question refers to the formation of GO and rGO via the chemical oxidation of graphite and the reduction of highly oxidised GO. Materials obtained by each method were shown to have differing characteristics for adsorption, oxidation of benzyl alcohol and electrical conductivity, despite being analogous in oxygen content. While there is a degree of overlap with our work with regards to level of oxidation, there was only one method of chemical oxidation used (Hummers). In the current work and the statement in question, we refer to the discrepancy between two different chemical oxidation methods (Hummers and Hofmann). We have referenced the paper and note the interesting results in the revised manuscript. However, we do not consider this earlier paper decreases the significance of our finding that the Hummers method produces inferior catalysts to those produced by Hofmann oxidation. This paper is also similar in principle to one by Liang et al. to which we refer on the oxidative dehydrogenation of isobutane.

(2) It appears that the epoxide products were characterized using gas chromatography. Were standards used to guide compound identification? The text indicates that multiple products form during the reaction (what is an "allylic product"?) and thus gas chromatography is insufficient because potential isomers may not be resolved. A secondary technique, such as NMR spectroscopy, should be employed to support the conclusions. Also, how were the gas chromatography signals quantified? Correction factors and/or calibration curves are typically required.

Response

The major products for this reaction are well known due to our previous work on the gold catalysed epoxidation of dec-1-ene (Gupta, U. N. et al. Solvent-free aerobic epoxidation of dec-1-ene using gold/graphite as a catalyst. Catal. Lett. 145, 689-696 (2015)). In all cases products were identified using GC-MS and confirmed using analytical standards. Reaction products were quantified using calibration with response factor obtained from standards of the identified compounds. The major product in all cases is the epoxide. Allylic products form a large part of the other major products. In the case of dec-1-ene these are dec-1-en-3-ol, dec-1-en-3-one, and dec-2-en-1-ol. Major products also include 1,2-decanediol and cracked acid and aldehyde products (C10, C9, C8). All products are clearly resolved using the optimised GC-method using a polar wax column. There are many unknown minor products which are assigned an averaged response factor. Selectivity to those compounds is, however, lower than 10%. For the purpose of this study focus was given to the selectivity towards the epoxide; however, in figure 1 combined allylic products were included. This is simply to elucidate the change over time from allylic oxidation to epoxidation. This has been discussed in our previous referenced work on the gold catalysed epoxidation of dec-1-ene. More information with regards to the analytical procedures of compounds identification is now provided in the paper.

(3) Since XPS data are not necessarily representative of the bulk sample, the GO products should also be analyzed using elemental analysis.

Response

We consider that XPS would give a good representation of oxygen and carbon compositions due to the layered structure of the material. It is the surface oxygen that we wish to characterise as this is what will be involved in the catalysis. Sulfur would likely be more prominent in the bulk due to the strong intercalation qualities. Inorganic sulfur was analysed by ion chromatography of the washings, while the presence of organosulfate was analysed by TGA-MS. Both of these methods should be sufficient to gain a good representation of the bulk. While elemental analysis of GO materials could provide further information, analysis of the bulk we do not consider is helpful in relation to this catalytic application and this is why we have used XPS for the surface analysis.

(4) Figure 1: Which catalyst was used to collect these data? Where are the data for the other catalysts? A table that summarizes key results would be informative.

Response

The data was obtained using GO-HO2O catalyst. Time online analysis was not conducted on all samples due to the time consuming nature of the experiment. These data were obtained to confirm the catalytic and non-stoichiometric nature of the reaction.

Minor Issues

(5) Line 108: "thermogravimetric" is spelled incorrectly.

Response

Word is corrected

(6) Many sections of the Results and Discussion section are obfuscated and long. The paragraph that begins on row 163 is 50 lines long! I suggest that the authors distill and clarify the content of their manuscript so that it is more accessible to a general audience.

Response

We thank the referee for this useful comment. The authors fully agree that several section of the manuscript require revision and changes. This issue has been addressed in the revised manuscript.

Response to reviewer #2

The paper by Hutchings and coworkers is well-written and of great impact. They have demonstrated that GO materials are active catalysts for aerobic epoxidation of aliphatic alkenes and elegantly unraveled details of the optimum level of carbocatalyst oxidation. Their finding that oxidation level

typically obtained in the permanagnate-based method is far from optimal will have impact on graphene preparation and this result may also open up new promising areas for oxidation of chemical feedstocks. I only have a few minor comments:

1. A reaction scheme showing the epoxidation performed in the study should be added.

Response

We agree and a representative scheme has been added.

2. A graphic showing sequential washing steps and the relative catalyst performance should be added.

Response

A Flowchart has been added. The authors agree that such change will add clarity to the findings.

3. Epoxidation conditions should be added in the discussion in the main manuscript.

Response

Epoxidation conditions have been added.

4. A relevant paper on nitronium ion-enabled synthesis of graphene materials should be cited (JACS 2012 134 5850).

Response

We have cited this paper as a potential method for production of sulfur-free catalysts. However, this paper describes materials specifically with a low oxygen composition. Therefore, although the materials would be sulfur-free, they would lack the level of oxygen required for activity, similar to those mentioned in the current paper produced by pyrolysis of sugars.

5. What is the catalyst performance in styrene or allylic alcohol epoxidation?

Response

These substrates have not been tested. Graphene related materials have been widely studied for oxidation reactions, however, the focus of this work was for the previously unreported epoxidation of terminal, linear alkenes. It would be interesting to assess the claims in this paper and the influence of sulfur on the epoxidation of internal alkenes. Our follow up study very much intends to address this question, however it was not the focus of this work. Our previous work on internal linear and cyclic alkenes would suggest the catalysts are active for these too in the absence of initiators.

6. Can the authors comment if they have tried the same approach for hydrocarbon, alcohol or amine oxidation?

Response

The catalysts were not tested for other hydrocarbon or amine oxidation. The catalysts were initially tested for benzyl alcohol oxidation using the work by Bielawski et al. as a reference. In both studies, no effect was observed between different oxidation methods. Also there was no benefit to using lower levels of oxidation due to sulfur not being a poison for benzyl alcohol oxidation as it appears to be in epoxidation of terminal linear alkenes.

All in all, this is a good piece of paper and I strongly recommend publication.

Response

We thank the referee for this strong support.

REVIEWERS' COMMENTS:

Reviewer #1 (Remarks to the Author):

I am satisfied with the authors' responses to the queries that were raised during the last round of review as well as the changes that were made to the manuscript. In my opinion, a stronger paper has emerged and is now appears to be ready for publication.

Response to final referee comments

Reviewer #1 (Remarks to the Author):

I am satisfied with the authors' responses to the queries that were raised during the last round of review as well as the changes that were made to the manuscript. In my opinion, a stronger paper has emerged and is now appears to be ready for publication.

Response

We thank the referee for these comments